# BMP2 Diminishes Angiotensin II-Induced Atrial Fibrillation by Inhibiting NLRP3 Inflammasome Signaling in Atrial Fibroblasts

**DOI:** 10.3390/biom14091053

**Published:** 2024-08-25

**Authors:** Yue Yuan, Hang Zhang, Erwen Xia, Xinbo Zhao, Qiang Gao, Hongyuan Mu, Xingzuo Liu, Yuanye Tian, Lei Liu, Qiuling Shen, Li Sheng

**Affiliations:** 1Department of Cardiology, The First Affiliated Hospital, Harbin Medical University, Harbin 150001, China; 601754@hrbmu.edu.cn (Y.Y.); 2022020679@hrbmu.edu.cn (H.Z.); mszxb5266288@163.com (X.Z.); 202101203@hrbmu.edu.cn (Q.G.); muhongyuan1994@163.com (H.M.); 2018152232@hrbmu.edu.cn (X.L.); 2023020939@hrbmu.edu.cn (Y.T.); 601896@hrbmu.edu.cn (L.L.); 813132@hrbmu.edu.cn (Q.S.); 2Department of General Medicine, The First Affiliated Hospital, Harbin Medical University, Harbin 150001, China; 2023020962@hrbmu.edu.cn

**Keywords:** atrial fibrillation, bone morphogenic protein 2, atrial fibrosis, inflammasome signaling

## Abstract

Atrial fibrillation (AF) is the most common sustained arrhythmia to affect 1% of the global population and increases with age. Atrial fibrosis is a crucial substrate for promoting structural remodeling to cause atrial arrhythmogenesis. Bone morphogenic protein 2 (BMP2) has been reported to be involved in cardiac fibrogenesis. However, its role in modulating atrial fibrosis to affect AF development remains unknown. Our study aimed to investigate the expression of BMP2 under different AF conditions and the effect of BMP2 on the progression of atrial fibrosis using an angiotensin II (Ang II) rat model and an ex vivo cardiac fibroblast model. The qRT-PCR and Western blot assay showed increased *BMP2* mRNA and protein levels in the atria of chronic AF patients and the right atria of a tachypacing rabbit model. In contrast, the levels of BMP2 receptor mRNA were comparable. The AF incidence of the Ang II rat was higher than that of a control rat, which was reduced by BMP2 treatment. Masson staining demonstrated an anti-fibrogenic impact on BMP2-subjected rat atria compared to only Ang II-treated rat atria. RNA-sequencing indicated the potential function of blocking NLRP3-associted inflammasome activation in BMP2-treated rat atrial tissues. In vitro, transfecting BMP2 shRNA into neonatal rat atrial fibroblasts upregulated the mRNA levels of *NLRP3/Caspase-1/p20/ASC* and the secretion of IL-1β and IL-6. In contrast, recombinant BMP2 protein attenuated the increased levels of the NLRP3 inflammasome pathway induced by Ang II. In summary, BMP2 opposes atrial fibrosis to alleviate AF susceptibility by inhibiting the activation of the inflammasome in atrial fibroblasts.

## 1. Introduction

Atrial fibrillation (AF), as one of the most common arrhythmia diseases, is associated with 1.5- to 2-fold increased risk of death and adverse outcomes such as heart failure, stroke and chronic kidney disease [1,2,3,4]. The AF incidence progressively augments each decade after 40 years old of age [4,5]. It was estimated that the global prevalence of AF already reached to 50 million in 2020 which resulted in a substantial public health burden [6,7]. Though there are multiple approaches including anti-arrhythmia drugs and catheter ablation to alleviate the occurrence and maintenance and restore heart rhythm, a huge number of patients are suffering from recurrency and adverse outcomes [8,9]. Emerging evidence have recognized AF as a progressive disease that emphasizes different management at different stages of AF [9]. Therefore, it is urgent to deeply explore the arrhythmogenic mechanism and develop potential therapeutic targets.

Atrial fibrosis is the most remarkable substrate for atrial structural remodeling, which can lead to slowed conduction velocity, pointing to re-entry to promote AF development [10,11]. The phenotype of cardiac fibroblasts (CFs) turned into myofibroblasts is the pivotal pathophysiologic change to secrete effective factors for collagen deposition accumulating in the atria [12]. Previous studies have shown that multiple pathways are involved in the activation of atrial fibrosis, such as the transforming growth factor (TGF)-β, Smad, and bone morphogenetic protein (BMP) families [13,14,15]. BMPs, as crucial members of the TGF-β superfamily, have been proven to exhibit anti-fibrogenic function in multiple organs [16,17]. Among these, BMP2 has been proven to suppress TGF-induced renal fibrogenesis signals in renal interstitial fibroblast cells and the differentiation and cell migration of lung fibroblasts [17,18]. Emerging evidence supports the notion that BMP2 exhibits a critical role in the regulation of bone marrow stromal cells into myofibroblasts [19]. Recently, BMP2 expression has been confirmed to localize in both cardiomyocytes and interstitial fibroblasts. BMP2 is involved in the formation of cardiac septa and angiogenesis by facilitating endothelial motility and invasion, which is crucial for the development of the adult heart [20,21]. Also, BMP2 has been proven to exert cytoprotective effects on cardiomyocytes via reducing inflammation in a model of non-reperfused myocardial infarction [22]. 

Recent studies have also suggested that BMPs play a key role in cardiovascular diseases, especially in regulating cardiac fibrosis. For example, BMP9 and BMP6 are identified to limit pathologic cardiac fibrosis and improve heart function in a transverse aortic constriction and myocardial infarction mouse model [23,24]. Moreover, BMP9 can protect against neonatal hypoxia-induced bronchopulmonary dysplasia by attenuating pro-inflammatory cytokine release [25]. Previous meta-analysis from bulk RNA-sequencing indicated that BMP2 upregulation in the atria of AF patients compared to sinus rhythm (SR) individuals [26], but its pathophysiological role in AF development and the potential molecular mechanisms are poorly understood. 

In the current study, we aimed to confirm the expression of BMP2 mRNA and protein and its receptor in different AF models. We also addressed its role in the occurrence and maintenance of atrial arrhythmia and clarified its effect on atrial fibrosis in angiotensin II (Ang II)-induced AF rat models. Furthermore, we tried to explain the response of rat neonatal atrial fibroblasts when exposed to recombinant BMP2 protein and the signaling pathway related to atrial fibrosis progression.

## 2. Methods

### 2.1. Patient Study

All individuals enrolled in this study were recruited from the Department of Cardiology, the First Affiliated Hospital in Harbin Medical University, as a single institution from September 2020 to June 2023. This study has been approved and registered by the Ethics Committee of the First Affiliated Hospital of Harbin Medical University (No. 201868). Informed consent was obtained from all participants. The AF group contained paroxysmal and chronic persistent AF patients but no permanent AF. Controls were age- and sex-matched with the AF group who had no AV block, atrial, or ventricular arrhythmia among consecutive participants in the same period. Exclusion criteria were the following: blood disease, chronic infection, cancer, severe chronic kidney diseases (GFR < 30 mL/min), heart transplant, coronary artery bypass surgery, acute heart failure, ischemia with non-obstructive coronary arteries (INOCA), including microvascular dysfunction and coronary spasm, cardiomyopathy (arrhythmogenic right ventricular dysplasia, hypertrophic cardiomyopathy, and dilated cardiomyopathy and takotsubo cardiomyopathy), pericarditis, and myocarditis. The plasma and atria samples of patients were collected from volunteers who were given the informed consent form. 

### 2.2. Animal Study

Eight- to ten-week-old male and female SD rats were purchased from Beijing Vital River Laboratory Animal Technology Co., Ltd., Beijing, China. All animal experiments and procedures were approved by the Institutional Animal Care and Use Committee of the First Affiliated Hospital of Harbin Medical University (No. 201868). The rats were housed in a temperature-controlled room (22–25 °C) with a 12 h light and dark cycle. The animals were randomly divided into control, Ang II (1 mg/kg/day) [27] for 2 weeks due to their age, comparable body weight, and unbiased gender, and Ang II rats were given saline or recombinant human BMP2 (70 μg/kg/day) injections for another 2 weeks [17]. Cardiac function and ventricular diameter were evaluated by Vevo 2100 Ultrasound. The pacing-induced AF susceptibility of rats was measured as described previously in detail [28]. AF was defined as more than 1 s of irregular atrial electrograms with irregular ventricular response. AF duration was defined as the mean AF episodes within 60 s in each animal. Ang II and recombinant human BMP2 were purchased from R&D Systems, Inc. (Minneapolis, MN, USA).

### 2.3. Cell Culture

Primary neonatal atrial cardiac fibroblasts (ACFs) were isolated from the atria of a 1–3 days-old SD rat, which had the species studied in vivo. The fresh neonatal rat atria were harvested, quickly minced, and digested with Liberase TH (Roche, Basel, Switzerland) for 30 min. After being filtered through a 100-μm nylon cell strainer (BD Falcon, Franklin Lakes, NJ, USA), the collected cells were incubated in fresh culture medium (DMEM with 10% fetal bovine serum) at 37 °C and 5% CO_2_ for 2 h to separate ACFs from atrial myocytes via differential plating. The ACFs were then cultured separately in high-glucose DMEM containing 10% FBS and 1% penicillin and streptomycin and were used in the following experiments after obtaining a 90% confluency.

### 2.4. mRNA Sequence

Rat atria were obtained, and total RNA were extracted and stored at −80 °C. mRNA sequencing was performed by a commercial service at Guangzhou RiboBio Co., Ltd. (Guangzhou, China) with the Illumina HiSeq 3000. Differential expression analysis was defined according to a fold-change (FC) ≥ 1.5 and *p* value < 0.05. The bioinformatic analysis was performed through Gene Ontology (GO) categories and the Kyoto Encyclopedia of Genes and Genomes (KEGG) database. 

### 2.5. qRT-PCR

Total RNAs of human and rat atria were extracted by a total RNA kit (Axygen, Union City, CA, USA), and cDNAs were reverse-transcribed by a high-capacity cDNA reverse transcription kit (Tyobo, Osaka, Japan). Quantitative real-time PCR was detected via SYBR Green and ABI PRISM 7500 Sequence Detection System (Applied Biosystems, Waltham, MA, USA). The primer sequences were designed according to the cDNA sequences in the GenBank database using Primer Express 3.0 software. *GAPDH* was used as the reference gene.

### 2.6. Western Blot

The whole protein lysis of atria was extracted, resolved on 8–12% SDS-PAGE, and transferred to a 0.45 μm polyvinylidene fluoride membrane. The membrane was blocked with 5% BSA for 1 h at room temperature, then incubated with primary antibodies overnight at 4 °C. The membrane was incubated with secondary antibodies for an hour and washed. Expressions of proteins were analyzed by the Bio-Rad system via the ECL kit (Thermo, Waltham, MA, USA). All detailed information on primary and secondary antibodies was provided in Appendix A.

### 2.7. Histology

Rat atria and atrial fibroblasts were fixed in 4% paraformaldehyde. H&E and Masson’s trichrome staining were performed as in the previous study [29].

### 2.8. Elisa Assay

Serum levels of cytokines, including IL-1β, IL-6, and TNF-α, were detected by Elisa kits according to the manufacturer’s instructions. The information for all kits was as follows: IL-1β (Elabscience, E-EL-M0037c), IL-6 (Elabscience, E-EL-M0044c), and TNF-α (Elabscience, E-MSEL-M0002).

### 2.9. Statistical Analysis

Quantitative data were shown as mean ± SEM. An unpaired Student’s *t* test was applied for comparisons in two groups. Multiple group comparisons were assessed through one-way ANOVA and Tukey’s multiple comparison tests. A two-way ANOVA analysis with multiple group comparisons was used to evaluate two or more main effect factors. Mann–Whitney and Kruskal–Wallis tests were used to compare the data, which showed that it was not normally distributed. Fisher’s exact tests were used to compare categorical data. *p* value < 0.05 was considered statistically significant.

## 3. Results

### 3.1. BMP2 Expression Upregulated in Different AF Model

In order to confirm the expression of BMP2 in different AF models, we measured the mRNA and protein levels in human atria samples and the right atria tachypacing (RAP) rabbit model. We harvested both right atria from patients with either sinus rhythm or AF episodes who underwent mitral valve transcription surgery. There were no significant differences in basic characteristics between the two groups of patients (Appendix A). qRT-PCR and Western blot showed increased BMP2 mRNA and protein in AF compared to SR patients (Figure 1A–C). We also found upregulation of BMP2 mRNA and protein in the atrial tissues of RAP rabbits compared with sham rabbits (Figure 1D–F). However, there was no change in BMP2 receptor mRNA in either SR or AF patient atria, consistent with sham and RAP rabbit atria (Figure 1G,H). These data suggest that BMP2 expression, not its receptor, increases in different AF models.

### 3.2. BMP2 Attenuated Angiotensin II Induced-AF Inducibility via Preventing Atrial Fibrosis In Vivo

To investigate the role of BMP2 protein in AF pathogenesis, we established an Ang II-induced AF rat model in which rats were randomly given saline or BMP2 intraperitoneal injections for 2 weeks (Figure 2A). To determine whether Ang II affects cardiac electrophysiology, we tested ECG parameters in three groups of rats. Heart rates, P wave, PR interval, QRS, and QT interval were comparable in all rats (Appendix A). A programmed intracardiac stimulation (PES) study was performed to test pacing-induced AF incidence. We found a higher AF inducibility of Ang II rats than control rats, but BMP2 treatment attenuated Ang II-induced AF incidence (Figure 2B,C). AF duration was shortened in BMP2-treated rats compared to Ang II rats (Figure 2D). To exclude that the increased AF inducibility of rats is caused by ventricular dysfunction, we used echocardiography to test the ventricle contractility. There were no significant changes in left ventricle eject fraction and left ventricular diameters in three groups of rats (Figure 2E,F). HE staining showed a trend of accumulated inflammatory cells infiltration in Ang II rat atria, which was reversed in BMP2 subjected rats (Figure 2G). We also evaluated atrial fibrosis level by Masson staining and qRT-PCR assay. We found more fibrosis in Ang II group compared to control rats, which was rescued by BMP2 treatment (Figure 2H, Appendix A). Then, we performed a qRT-PCR assay to measure BMP2 receptor mRNA expression in rat atria. The results showed an unaltered level of BMP2R in Ang II + BMP2 rat atria compared to Ang II rat atria (Appendix A). We also observed an activation of p-SMAD1/5/8 signaling via Western blot in the Ang II + BMP2 group compared to the Ang II group (Appendix A). These results demonstrate that BMP2 treatment attenuates Ang II-associated AF susceptibility and profibrotic atrial remodeling.

### 3.3. RNA-Sequence Identify the Negative Role of BMP2 in Rat Atrial Inflammation 

To determine how BMP2 alleviates atrial electrophysiology, we performed RNA-seq in Ang II and Ang II + BMP2 rat atria. There were 147 differential genes between two groups of rat atria (Figure 3A). A qRT-PCR test was used to verify five of the most differential genes related to fibrosis and inflammation in the rat atrial samples (Appendix A). GO and KEGG analysis indicated that BMP2 can inhibit NOD-, LRR-, and pyrin domain-containing protein 3 (NLRP3)-mediated inflammation in rat atria (Figure 3B). Then, we performed a qRT-PCR assay to verify the mRNA levels of the NLRP3 inflammasome pathway among all three groups of rats. The results showed upregulation of *NLRP3*, *Caspase1*, *FL-GSDMD* (full-length gasdermin D), *NT-GSDMD* (N terminal gasdermin D) and *ASC* mRNAs in the Ang II-associated rat compared to the control rat (Figure 3C). In contrast, the Ang II + BMP2 rat showed decreased mRNA expression in the atria compared to the Ang II-treated rat (Figure 3C). Moreover, we tested serum levels of IL-1β, IL-6, and TNF-α which are related to the development of atrial arrhythmia. Compared with control rats, the levels of serum IL-1β, IL-6, and TNF-α enhanced in Ang II rats (Figure 3D–F). While Ang II + BMP2 rats exhibited lower levels of serum IL-1β, IL-6, and TNF-α than Ang II rats (Figure 3D–F). These data suggest that BMP2 reduces the activation of the NLRP3 inflammasome pathway in Ang II-treated rat atria.

### 3.4. Knockdown BMP2 Enhances NLRP3-Dependent Inflammasome Activation in Rat Atrial Fibroblast

To clarify the role of BMP2 by regulating the NLRP3 inflammasome pathway to affect the phenotype of atrial fibroblasts, we isolated neonatal rat atrial fibroblasts and knocked down BMP2 by siRNA. qRT-PCR data confirmed an inhibited expression of *BMP2* mRNA in atrial fibroblasts (Figure 4A). Then, we found the upregulation of *NLRP3*, *Caspase1*, *FL-GSDMD,* and *NT-GSDMD* mRNA levels in si-BMP2 rat atrial fibroblasts while maintaining unchanged *ASC* mRNA (Figure 4B). The EdU assay suggested increased proliferation of the si-BMP2 group compared to negative control (NC) atrial fibroblasts (Figure 4C,D). αSMA staining was performed to detect the differentiated viability of rat atrial fibroblasts. The data showed an enhanced αSMA signal of atrial fibroblast in si-BMP2 group compared to the control group (Figure 4E,F). In addition, we found increased mRNA levels of *Collagen 1* and *αSMA* in the si-BMP2 group when compared to NC atrial fibroblasts (Figure 4G). These results demonstrate a profibrogenic effect of decreased BMP2 on atrial fibroblasts via activating NLRP3-mediated inflammasomes. 

### 3.5. BMP2 Prevents Ang II-Induced NLRP3 Inflammasome Signaling in Neonatal Rat Atrial Fibroblasts

Previous work has shown that BMP2 inhibits NLRP3 inflammasome activation during the development of type 2 diabetes mellitus-induced cardiomyocyte injury [30]. To evaluate whether BMP2 exerts a negative role for the NLRP3 inflammasome pathway in Ang II-treated atrial fibroblasts, we measured mRNA levels of NLRP3 signaling in control, Ang II, and Ang II + BMP2 neonatal rat atrial fibroblasts. qRT-PCR results showed increased mRNA levels of *NLRP3*, *Caspase1*, *FL-GSDMD*, *NT-GSDMD,* and *ASC* in Ang II-treated atrial fibroblasts (Figure 5A). Moreover, the mRNA levels of *NLRP3*, *Caspase1*, *FL-GSDMD*, *NT-GSDMD,* and *ASC* were mitigated by BMP2 (Figure 5A). Then, we performed an EdU assay to observe the proliferation of different groups of atrial fibroblasts. The data indicated that BMP2 diminished the Ang II-induced ratio of EdU^+^ atrial fibroblasts (Figure 5B,C). In addition, αSMA staining was used to detect rat atrial fibroblast differentiation. As shown, Ang II significantly increased αSMA signal of atrial fibroblast, which was reversed by BMP2 treatment (Figure 5D,E). The mRNA levels of *Collagen 1* and *αSMA* in Ang II-treated atrial fibroblasts increased compared to Ctl group, which were downregulated by BMP2 (Figure 5F). These results establish that BMP2 decreases the activated NLRP3 inflammasome and profibrogenic phenotype in Ang II-treated rat atrial fibroblasts.

## 4. Discussion

Angiotensin II can induce systemic inflammation to result in atrial fibrosis, which, as an arrhythmia substrate, increases the risk factor for AF. BMP family molecules, such as BMP2, are frequently associated with organ fibrosis. Recently, emerging evidence suggests that BMP2 is involved in the activity of inflammation to modulate the fibrogenic process [30,31]. However, whether BMP2 affects NLRP3-mediated inflammatory signaling in atrial fibroblasts to improve AF development. In the current study, we reported that: (i) the upregulation of BMP2 in the atria of both AF patients and animal models, (ii) BMP2 inhibits the NLRP3 inflammasome pathway and atrial fibrosis to alleviate the inducibility of Ang-II-induced AF, (iii) BMP2 exerts an anti-inflammation and anti-fibrogenic role in atrial fibroblasts to ameliorate Ang-II-related fibrogenesis. Our data point out that BMP2 is a potential inhibitor to diminish the angiotensin II-associated NLRP3 inflammasome pathway in atrial fibroblasts to prevent AF pathogenesis.

BMPs can regulate proper cardiac development by coordinating atrioventricular canal morphogenesis [32]. BMP2, BMP4, and BMP7 are highly expressed in the cardiomyocytes, and have distinct functions [33]. Among them, BMP2 has an essential pleiotropic and multifunctional role, having a wide range of actions in different types of cells [34]. In the study of myocardial infarction, BMP2 performs early induction of protein expression in both cardiomyocytes and interstitial fibroblasts [35,36]. However, its role in the occurrence and maintenance of atrial arrhythmia remains unknown. Previous data only show the upregulation of BMP2 in the atrial tissues of AF patients [26]. The current results also confirm the increased expression of BMP2 mRNA and protein in the atria of AF patients and RAP rabbits. Furthermore, our findings identify that BMP2 attenuates the incidence of Ang-II-stimulated AF and abbreviates AF durations in rats. These data point to the beneficial role of BMP2 in preventing the development of AF in vivo. In addition, we found that the mRNA level of BMP2R in the atria of AF patients and RAP rabbits was unchanged when compared to controls. Several studies have confirmed elevated BMP2 receptor levels induced by IL-6 signaling can activate SMAD1/5/8 and dampen the SMAD 2/3 pathway in idiopathic pulmonary fibrosis [20]. Though our data showed comparable BMP2R mRNA in the AF model, further studies will be needed to explore the role of the BMP2 receptor and signaling pathway in the pathogenesis of atrial fibrosis.

Accumulated evidence supports that the function of endogenous BMP2 in regulating fibroblast phenotypes is completely opposite in different tissues. Much like TGF-β1, BMP2 promotes the transformation of mesenchymal stem cells into a profibrotic phenotype for secreting factors in the osteo-organoid maturation process [37]. Nevertheless, it is shown that BMP2 reverses the fibrogenic function of TGF-β1 in pancreatic stellate cells via the Smad1 signaling pathway [38]. However, there is less data to clarify whether BMP2 has an anti-fibrosis function to mitigate re-entry-related AF substrates to prevent AF arrhythmogenesis. Our results suggest that BMP2 treatment reduces Ang-II-induced atrial fibrosis to inhibit AF development in vivo. Moreover, BMP2 ameliorates stress-stimulated profibrogenic phenotype transforming of primary rat atrial fibroblasts in vitro. Our data are consistent with other studies in myocardial infarction-associated heart failure, indicating that BMP2 could exert a negative role of cardiac fibrosis in different cardiovascular diseases [39]. 

It is reported that the activation of the NLRP3 inflammasome pathway in cardiac fibroblasts can cause Ang-II-induced cardiovascular diseases, including atrial arrhythmia [40,41]. Growing evidence has demonstrated that BMP2 is involved in the systematic inflammatory response to various heart diseases. Enhanced BMP2 expression is found in the atherosclerotic lesions, colocalized into the inflammatory and oxidative stress area [42]. Overexpression of BMP2 suppresses NLRP3-mediated pyroptosis in diabetic hearts [30]. Consistent with these results, our sequencing data provided compelling evidence that BMP2 inhibits the activity of NLRP3 inflammation in atrial fibroblasts to diminish AF susceptibility. Previously, BMP2 was proven to antagonize BMP4-associated cardiomyocyte apoptosis, and BMP2 alone does not contribute to apoptosis [43]. Our present results demonstrated that BMP2 alone is necessary and sufficient to protect atrial fibroblasts from differentiation, proliferation, and cytokine release stimulated by Ang-II. While whether immune cells such as neutrophils and macrophages, are essential effector cells involved in NLRP3 inflammasome signaling, which is targeted by BMP2, remains unknown. The molecular mechanisms should be explored in future studies.

Our study has some limitations. The enrolled AF patients had other potential risk factors for AF, including diabetes and hypertension. They also received treatment with anti-diabetic and anti-hypertensive drugs, which might have an impact on atrial structural remodeling. According to the clinical techniques, we only collected right atrial tissues for BMP2 mRNA and protein tests. These data should be considered to carefully interpret the human data. In addition, our study has a small number of human AF atrial samples to examine BMP2 and its receptor expression. Further studies with larger sample sizes are needed to confirm the change in BMP2 in AF patients and clarify its correlation with AF progress. Although we observed that BMP2 decreased the activity of NLRP3 inflammasomes in atrial fibroblasts, the precise molecular mechanisms are unclear. Several studies have indicated that BMP2 is associated with the production of reactive oxygen species (ROS), which can directly activate the NLRP3 inflammasome pathway [44,45]. Therefore, the accurate molecular relationship between BMP2 and NLRP3 signaling and the phenotype transformation of atrial fibroblasts should be addressed in the following studies to confirm the sources of ROS production. Moreover, it is needed to examine the role of BMP2R on activated inflammation in atrial fibroblasts and provide insight into its potential mechanism related to the NLRP3 signaling pathway.

## 5. Conclusions

In summary, our work confirms that BMP2 exhibits an anti-inflammatory role in fibroblasts to prevent atrial fibrosis and reduce the susceptibility to angiotensin II-associated AF. The current data suggest that BMP2 might be considered a potential pharmacological target for the treatment of atrial arrhythmia.

## Figures and Tables

**Figure 1 biomolecules-14-01053-f001:**
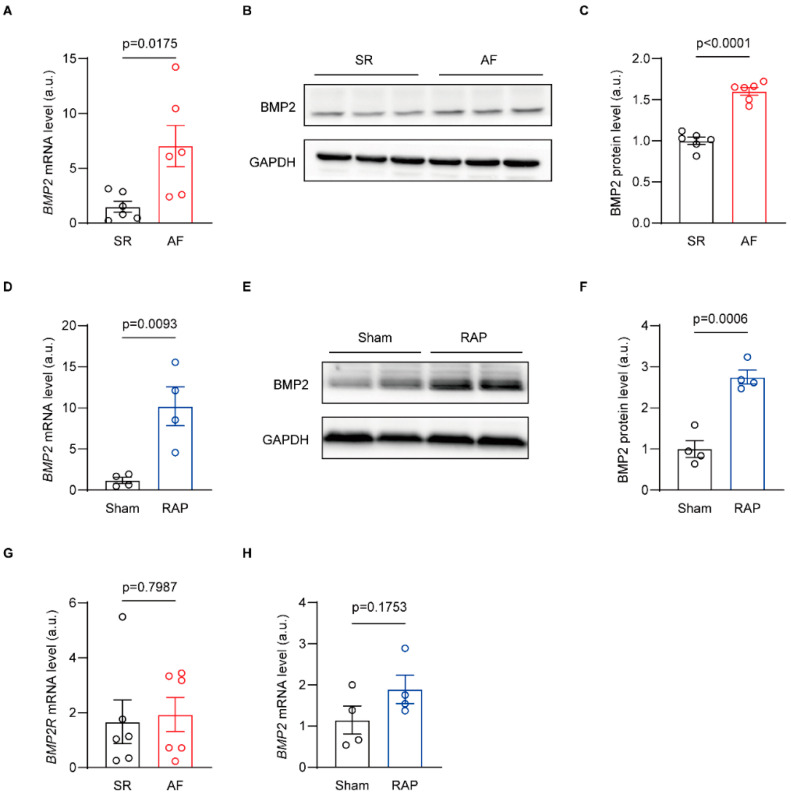
BMP2 expression increased in the atria of different AF models. (**A**) qRT-PCR-tested BMP2 mRNA level in SR and AF patient atria, n = 6/group. (**B**,**C**) Western blot tested BMP2 protein level in SR and AF patient atrial samples, n = 6/group. (**D**) qRT-PCR-tested *BMP2* mRNA level in sham and RAP atria, n = 4/group. (**E**,**F**) Western blot-tested BMP2 protein level in sham and RAP rabbit atria, n = 4/group. (**G**) BMP2 receptor mRNA level in SR and AF patient atria, n = 6/group. (**H**) qRT-PCR-tested BMP2 receptor mRNA level in sham and RAP rabbit atrial tissues, n = 4/group. SR, sinus rhythm; AF, atrial fibrillation; RAP, right atria tachypacing. The bar graph data are mean ± SEM with individual values. *p*-values are determined with unpaired Student’s *t*-test in (**A**,**C**,**D**,**F**,**G**,**H**).

**Figure 2 biomolecules-14-01053-f002:**
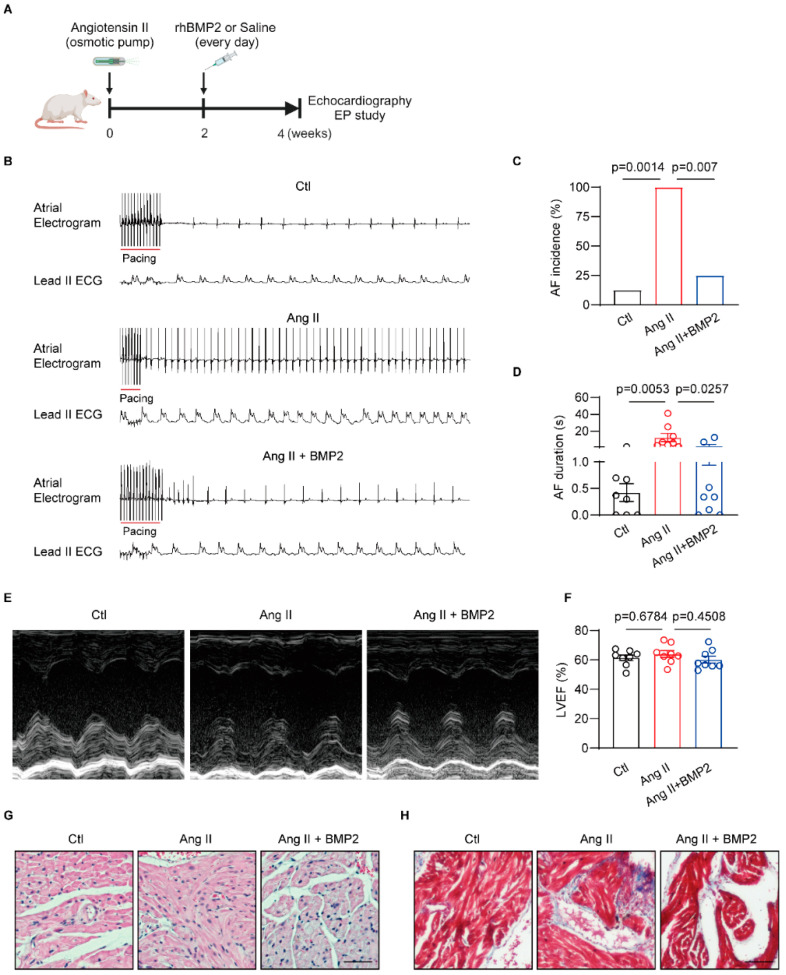
BMP2 alleviated Ang-II induced AF incidence and atrial fibrosis in rats. (**A**) The scheme of Ang II and recombinant human BMP2 treatment in rats, n = 8/group. (**B**) The representative surface ECG and atrial electrogram of Ctl, Ang II and Ang II + BMP2 rats. (**C**,**D**) The AF incidence and AF duration in Ctl, Ang II and Ang II + BMP2 rats, n = 8/group. (**E**) Representative LV M-mode echocardiograph images. (**F**) The measurement of left ventricular ejection fraction in Ctl, Ang II and Ang II + BMP2 rats, n = 8/group. (**G**) Representative HE staining images of the three groups of rat atria, n = 3/group. Scale bar: 100 μm. (**H**) Representative Masson staining images of in three groups of rats, n = 3/group. Scale bar: 100 μm. Ctl, control; angiotensin II, Ang II; LVEF, left ventricle ejection fraction; AF, atrial fibrillation. The bar graph data are mean ± SEM with individual values. *p*-values are determined with one-way ANOVA and Turkey’s multiple comparisons test in (**F**), Fisher’s exact test in (**C**), and Mann–Whitney test in (**D**).

**Figure 3 biomolecules-14-01053-f003:**
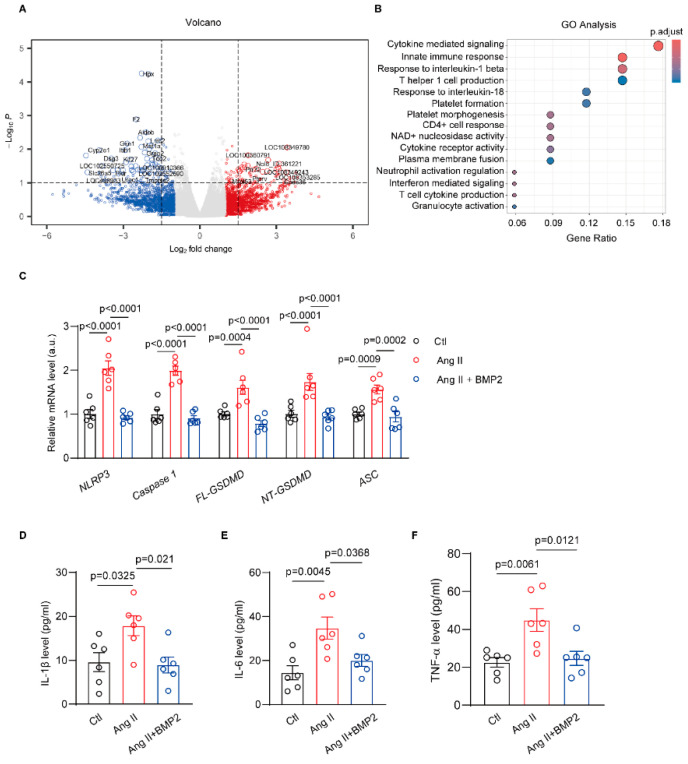
RNA-sequence identify the negative role of BMP2 in rat atrial inflammation. (**A**) The volcano plot to present the differential genes in Ang II and Ang II + BMP2 rat atria, n = 3/group. (**B**) GO analysis showing potential function of the related differential genes. (**C**) qRT-PCR assay verified the mRNA levels of *NLRP3*, *Caspase 1*, *FL-GSDMD*, *NT-GSDMD*, and *ASC* in three groups of rat atria, n = 6/group. (**D**–**F**) Elisa test showing the serum levels of IL-1β, IL-6, and TNF-α in sham, Ang II and Ang II + BMP2 rats, n = 6/group. Ctl, control; angiotensin II, Ang II. The bar graph data are mean ± SEM with individual values. *p*-values are determined with one-way ANOVA and Turkey’s multiple comparisons test in (**C**–**F**).

**Figure 4 biomolecules-14-01053-f004:**
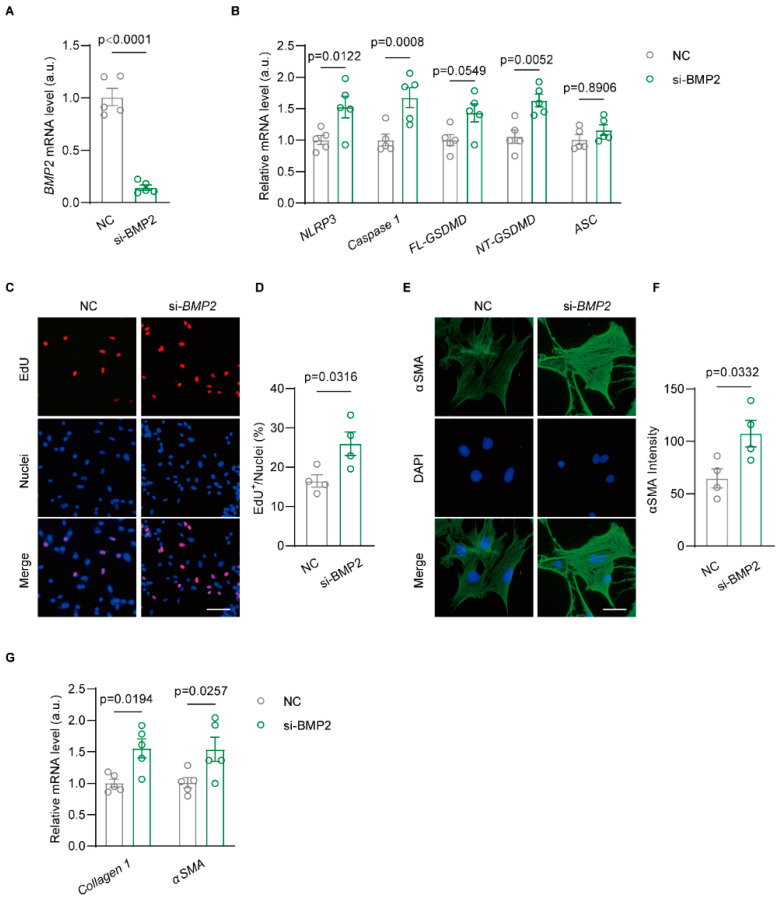
Knockdown BMP2 enhances NLRP3 inflammasome pathway in rat atrial fibroblasts. (**A**) qRT-PCR assay showed *BMP2* mRNA levels in NC and si-BMP2 rat atrial fibroblasts, n = 5/group. (**B**) qRT-PCR results verified the mRNA levels of *NLRP3*, *Caspase 1*, *FL-SDMD*, *NT-GSDMD* and *ASC* in NC and si-BMP2 ACFs, n = 5/group. (**C**,**D**) EdU testing the proliferation of NC and si-BMP2 ACFs, n = 4/group. Scale bar: 50 μm. (**E**,**F**) αSMA staining in NC and si-BMP2 ACFs, n = 4/group. Scale bar: 25 μm. (**G**) qRT-PCR results verified mRNA levels of *Collagen1* and *αSMA* in NC and si-BMP2 ACFs, n = 5/group. Negative control, NC and knockdown *BMP2*, si-BMP2. The bar graph data are mean ± SEM with individual values. *p*-values are determined with unpaired Student’s *t*-test in (**A**,**B**,**D**,**F**,**G**).

**Figure 5 biomolecules-14-01053-f005:**
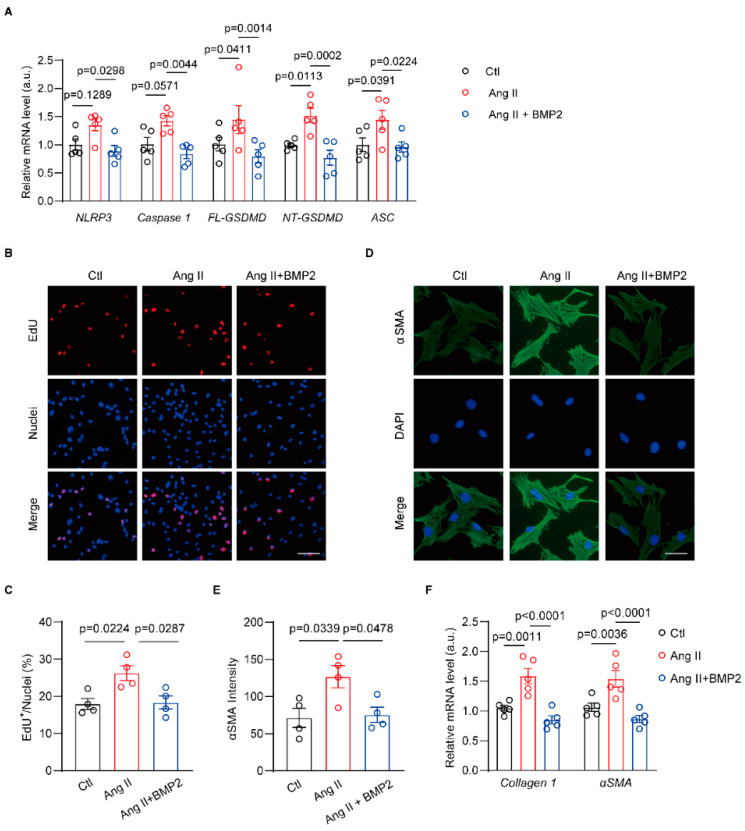
BMP2 inhibits Ang II-associated NLRP3 inflammasome activation in rat atrial fibroblasts. (**A**) qRT-PCR assay was performed to verify the mRNA levels of *NLRP3*, *Caspase 1*, *FL-SDMD*, *NT-GSDMD* and *ASC* in control (Ctl), Ang II and Ang II + BMP2 ACFs, n = 5/group. (**B**,**C**) EdU assay tested the proliferation of Ctl, Ang II and Ang II + BMP2 ACFs, n = 4/group. Scale bar: 50 μm. (**D**,**E**) αSMA staining in Ctl, Ang II and Ang II + BMP2 ACFs, n = 4/group. Scale bar: 25 μm. (**F**) qRT-PCR was performed to measure the mRNA levels of *Collagen1* and *αSMA* in Ctl, Ang II and Ang II + BMP2 ACFs, n = 5/group. Ctl, control; angiotensin II, Ang II. The bar graph data are mean ± SEM with individual values. *p*-values are determined with unpaired Student’s *t*-test in (**C**,**E**), one-way ANOVA and Turkey’s multiple comparisons test in (**A**,**F**).

## Data Availability

All data included in this paper will be shared by the corresponding author upon reasonable request.

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
