# Peer review of "BMP2 Diminishes Angiotensin II-Induced Atrial Fibrillation by Inhibiting NLRP3 Inflammasome Signaling in Atrial Fibroblasts"

_biomolecules, 2024, doi:10.3390/biom14091053_

Round 1
Reviewer 1 Report
Comments and Suggestions for Authors
This study investigates the role of BMP2 in preventing Ang II-induced AF. Atrial fibrosis is a key factor in promoting atrial structural remodeling, which in turn contributes to arrhythmogenesis. Previous research has suggested that BMP2 is involved in fibrogenesis, but its specific role in the development of AF is not well understood.
The study aims to explore BMP2 expression under AF conditions and its effect on the progression of atrial fibrosis. The results could reveal new therapeutic targets for managing AF.
The introduction provides adequate context on AF and atrial fibrosis, and introduces the potential role of BMP2 in cardiac fibrogenesis. However, it might be useful to add a brief overview of the current knowledge on the role of BMP proteins in atrial fibrosis and the NLRP3 inflammasome, to better justify the study.
I suggest to provide more details on how the patients were selected and the exclusion criteria used.
I suggest to include a more detailed discussion of negative or unexpected results, if any.
The discussion is well-articulated and correctly interprets the study's results. It connects the findings to existing knowledge and suggests potential therapeutic implications of BMP2 in AF. However, the discussion could benefit from a more detailed section on the study's limitations and future research directions.
Comments on the Quality of English LanguageSome sentences are long and complex, a review can improve clarity and conciseness.
Author Response
Response: We would like to thank you for summarizing and highlighting the main findings of our manuscript. We also appreciate the insightful suggestions for all your comments. The point-to-point response is as below to provide detailed updates of the revised manuscript.
- The introduction provides adequate context on AF and atrial fibrosis and introduces the potential role of BMP2 in cardiac fibrogenesis. However, it might be useful to add a brief overview of the current knowledge on the role of BMP proteins in atrial fibrosis and the NLRP3 inflammasome, to better justify the study.
Response: We appreciate the reviewer’s comments. In this revision, we provide more background knowledge of BMP proteins such as BMP6 and BMP9 in the pathogenesis of cardiac fibrosis and its regulation of NLRP3-related cytokines release in other diseases. We added the contents related to this point in the introduction. (Page 2, Line 62-67)
‘Recent literatures have also suggested that BMPs plays a key role in cardiovascular diseases especial for regulating cardiac fibrosis. For example, BMP9 and BMP6 are identified to limit pathologic cardiac fibrosis and improve the heart function in transverse aortic constriction and myocardial infarction mouse model23,24. Moreover, BMP9 can protect against neonatal hypoxia-induced bronchopulmonary dysplasia by attenuating pro-inflammatory cytokine release25.’
- I suggest to provide more details on how the patients were selected and the exclusion criteria used.
Response: We would like to thank the reviewer for this suggestion. More detail of the inclusion and exclusion criteria were provided in the ‘Methods’ as below. (Page 2, Line 83-92)
‘AF group contained paroxysmal and chronic persistent AF patients, but no permanent AF. Controls were age- and sex-matched with AF group who had no AV block, atrial and ventricular arrhythmia among consecutive participants in the same period. Exclusion criteria were the following: blood disease, chronic infection, cancer, severe chronic kidney diseases (GFR<30ml/min), heart transplant, coronary artery bypass surgery, acute heart failure, ischemia with non-obstructive coronary arteries (INOCA) including microvascular dysfunction and coronary spasm, cardiomyopathy (arrhythmogenic right ventricular dysplasia, hypertrophic cardiomyopathy, and dilated cardiomyopathy and takotsubo cardiomyopathy), pericarditis and myocarditis.’
- I suggestto include a more detailed discussion of negative or unexpected results, if any.
Response: Thank you for this suggestion. Our current study had shown an unchanged level of BMP2 receptor mRNA in AF patients and animal model compared to controls (Figure 1G and H). In the current study, we did not explore the role of BMP2 receptor on the pathogenesis of atrial fibrosis in the AF development. Thus, we included the discussion to interpret this negative data. (Page 10, Line 317-323)
‘Besides, we found that the mRNA level of BMP2R in the atria of AF patients and RAP rabbits were unchanged when compared to controls. Several studies have confirmed elevated BMP2 receptor level induced by IL-6 signaling can activate SMAD1/5/8 and dampen SMAD 2/3 pathway in idiopathic pulmonary fibrosis (2016 AJPL). Though our data showed comparable BMP2R mRNA in AF model, followed studies will be needed to explore the role of BMP2 receptor and signaling pathway in the pathogenesis of atrial fibrosis.’
- However, the discussion could benefit from a more detailed section on the study's limitations and future research directions.
Response: Thank you for this excellent comment. Due to the timeframe of the revision, we were unable to examine the level of BMP2 protein in human AF atrial samples. We think that further studies with larger sample sizes are needed to confirm the change of BMP2 in AF patients. Besides, it is better to explore the change of SMAD pathway in AF and how activated NLRP3 signaling regulates SMAD pathway. We have updated the discussions accordingly.
Reviewer 2 Report
Comments and Suggestions for Authors
The study explores the role of bone morphogenic protein 2 (BMP2) in modulating atrial fibrosis and its potential therapeutic effects on atrial fibrillation (AF). The authors utilized both in vivo (Angiotensin II-induced AF rat model) and in vitro (neonatal rat atrial fibroblasts) models to elucidate the mechanisms by which BMP2 influences the NLRP3 inflammasome signaling pathway.
Strengths
- Comprehensive Approach: The study employs a combination of in vivo and in vitro models, providing a thorough investigation into BMP2's role in AF.
- Molecular Techniques: Use of qRT-PCR, Western blot, RNA sequencing, and ELISA assays to measure gene and protein expression levels, providing robust data on the molecular mechanisms involved.
- Clear Results: The findings indicate that BMP2 treatment reduces AF incidence and atrial fibrosis by inhibiting the NLRP3 inflammasome pathway, offering potential therapeutic insights.
- Detailed Methodology: The experimental procedures are well-detailed, allowing for reproducibility and verification by other researchers.
Weaknesses
- Clinical Relevance: While the study shows promising results in animal models, the translation of these findings to human clinical practice requires further investigation.
- Sample Size: The sample size for some experiments is relatively small, which might limit the statistical power and robustness of the conclusions.
- BMP2 Receptor Analysis: The study mentions that BMP2 receptor mRNA levels were comparable across different groups, but does not delve deeply into the functional implications of this finding.
- Potential Confounding Factors: The human study participants had various potential confounders (e.g., diabetes, hypertension, and concurrent medications) that could influence atrial structural remodeling and AF progression.
Detailed Comments and Suggestions
-
Introduction
- The introduction provides a solid background on AF and the role of atrial fibrosis. However, it would benefit from a more detailed discussion on previous studies linking BMP2 with cardiovascular diseases.
-
Methods
- Patient Study: More details on the inclusion and exclusion criteria would be beneficial. Additionally, a larger and more diverse patient cohort could strengthen the study's findings.
- Animal Study: The randomization process for the animal groups should be described in more detail. Also, considering the potential effects of sex differences in response to treatments, it would be valuable to specify the sex of the animals used.
- Cell Culture: The source of neonatal rat atrial fibroblasts and the rationale behind choosing this specific cell type should be clearly stated.
-
Results
- BMP2 Expression: The increase in BMP2 mRNA and protein levels in AF models is well-documented. Including additional data on BMP2 receptor functionality could provide more insight into the signaling pathways involved.
- Ang II-Induced AF Model: The results show that BMP2 treatment reduces AF incidence and atrial fibrosis. However, providing more quantitative data on fibrosis reduction and correlating it with BMP2 levels would strengthen this section.
- RNA Sequencing: The RNA-seq data is promising. It would be beneficial to validate a few key differentially expressed genes through independent methods such as qRT-PCR or Western blot.
-
Discussion
- The discussion aptly summarizes the findings and their implications. However, it should address potential limitations more thoroughly, such as the small sample size and the need for further validation in larger cohorts.
- The discussion on the potential clinical applications of BMP2 as a therapeutic target for AF is interesting. However, more emphasis on the challenges and future directions for translating these findings into clinical practice is needed.
-
Figures and Tables
- Figures are well-presented and provide clear visual representation of the data. Ensuring that all figures have comprehensive legends and are referred to appropriately in the text will improve clarity.
- Tables summarizing patient demographics and experimental conditions are useful. Adding a table with key RNA-seq findings could also be beneficial.
Conclusion
This study provides valuable insights into the role of BMP2 in reducing angiotensin II-induced atrial fibrillation by inhibiting NLRP3 inflammasome signaling. While the findings are promising, further studies with larger sample sizes and clinical trials are needed to confirm the therapeutic potential of BMP2 in AF. Addressing the mentioned weaknesses and suggestions will strengthen the manuscript and its contributions to the field.
Author Response
Response: We would like to thank you for summarizing and highlighting the main findings of our manuscript. We also appreciate the insightful suggestions for all your comments. The point-to-point response is as below to provide detailed updates of the revised manuscript.
- The introduction provides a solid background on AF and the role of atrial fibrosis. However, it would benefit from a more detailed discussion on previous studies linking BMP2 with cardiovascular diseases.
Response: We would like to thank you for this comment. In this revision, we provide more background knowledge of BMP2 and cardiovascular diseases. We added the contents related to this point in the introduction. (Page 2, Line 56-61)
‘Recently, BMP2 expression is confirmed to localize in both cardiomyocytes and interstitial fibroblasts. BMP2 is involved in the formation of cardiac septa and angiogenesis by facilitating endothelial motility and invasion which is crucial for the development of adult heart20, 21. Also, BMP2 is proved to exert cytoprotective effects on cardiomyocytes via reducing inflammation in a model of non-reperfused myocardial infarction22.
’
- Patient Study: More details on the inclusion and exclusion criteria would be beneficial. Additionally, a larger and more diverse patient cohort could strengthen the study's findings.
Response: Thank you for this brilliant suggestion. More detail of the inclusion and exclusion criteria were provided in the ‘Methods’ as below. (Page 2, Line 83-92)
‘AF group contained paroxysmal and chronic persistent AF patients, but no permanent AF. Controls were age- and sex-matched with AF group who had no AV block, atrial and ventricular arrhythmia among consecutive participants in the same time period. Exclusion criteria were the following: blood disease, chronic infection, cancer, severe chronic kidney diseases (GFR<30ml/min), heart transplant, coronary artery bypass surgery, acute heart failure, ischemia with non-obstructive coronary arteries (INOCA) including microvascular dysfunction and coronary spasm, cardiomyopathy (arrhythmogenic right ventricular dysplasia, hypertrophic cardiomyopathy, and dilated cardiomyopathy and takotsubo cardiomyopathy), pericarditis and myocarditis.’
3.Animal Study: The randomization process for the animal groups should be described in more detail. Also, considering the potential effects of sex differences in response to treatments, it would be valuable to specify the sex of the animals used.
Response: We would like to thank the reviewer’s suggestion. We provided the information for the randomization process of rats and the gender of animals. Please see the changes: (Page 3, Line 95-102)
‘8-10 weeks-old male and female SD rats were purchased from Beijing Vital River Laboratory Animal Technology Co., Ltd. All animal experiments and procedures were approved by the Institutional Animal Care and Use Committee of the First Affiliated Hospital of Harbin Medical University. The rats were housed in a temperature-controlled room (22-25°C) with 12-hours light and dark cycle. The animals were randomly divided into control, Ang II (1mg/kg/day)27 for 2 weeks due to the age, comparable body weight and unbiased gender, and Ang II rats were given with saline or recombinant human BMP2 (70μg/kg/day) injection for another 2 weeks17.’
4.Cell Culture: The source of neonatal rat atrial fibroblasts and the rationale behind choosing this specific cell type should be clearly stated.
Response: Thank you for the suggestion. We used 1-3 days new born SD rats which were the same specie as in vivo study to harvest fresh heart and separate the atria to collect primary atrial fibroblast. We only used the primary cells with no passage to keep the original phenotype. More detail was updated in the revision: (Page 3, Line 109-112)
‘Primary neonatal atrial cardiac fibroblasts (ACFs) were isolated from the atria of 1-3 days-old SD rat which had the species with in vivo study. The fresh neonatal rat atria were harvested, quickly minced, and digested with Liberase TH (Roche, Basel, Switzerland) for 30 min.’
5.BMP2 Expression: The increase in BMP2 mRNA and protein levels in AF models is well-documented. Including additional data on BMP2 receptor functionality could provide more insight into the signaling pathways involved.
Response: Thank you for this insightful comment. During this revision, we have performed qRT-PCR assay to measure the levels of BMP2 receptor in rat atria. There was no significant change of BMP2 receptor in Ang II subjected rat compared to Ang II+BMP2 rat (Supplemental Figure 2B). Besides, we observed an elevated level of P-SMAD1/5/8 in BMP2 treatment rat (Supplemental Figure 2C and D). These data suggested that BMP2 receptor could regulate SMAD1/5/8 pathway in the progress of atrial fibrosis. Due to the timeframe of the revision, we were unable to examine the function of BMP2 receptor in vivo. A future study is needed to further elucidate the exact role of BMP2 receptor in the promotion of atrial fibrosis and atrial arrhythmogenesis. We now revised the text in the ‘Result and Discussion’ section to reflect the information. (Page 10, Line 317-323)
‘Besides, we found that the mRNA level of BMP2R in the atria of AF patients and RAP rabbits were unchanged when compared to controls. Several studies have confirmed elevated BMP2 receptor level induced by IL-6 signaling can activate SMAD1/5/8 and dampen SMAD 2/3 pathway in idiopathic pulmonary fibrosis20. Though our data showed comparable BMP2R mRNA in AF model, followed studies will be needed to explore the role of BMP2 receptor and signaling pathway in the pathogenesis of atrial fibrosis.’
6.Ang II-Induced AF Model: The results show that BMP2 treatment reduces AF incidence and atrial fibrosis. However, providing more quantitative data on fibrosis reduction and correlating it with BMP2 levels would strengthen this section.
Response: We would like to thank you for the comments. We performed qRT-PCR assay to measure the mRNA levels of Collagen1 and aSMA in rat atria. The new results showed that increased BMP2 level attenuated fibrosis in Ang II-associated AF rats in Supplemental Fig.2A. These data indicated BMP2 can attenuate Ang II-induced atrial fibrosis. We updated this new data in the revision. Please see the changes. (Page 5, Line 193-195)
‘We also evaluated atrial fibrosis level by Masson staining and qRT-PCR assay. We found more fibrosis in Ang II group compared to control rats, which was rescued by BMP2 treatment (Fig.2H, Supplemental Fig.2A).’
- RNA Sequencing: The RNA-seq data is promising. It would be beneficial to validate a few key differentially expressed genes through independent methods such as qRT-PCR or Western blot.
Response: We would like to thank you this suggestion. During the revision, qRT-PCR was performed to verify the levels of most differential 5 genes related to the regulation of fibrosis and inflammation including Co1, IL-18, BMP10, Fgb and TIMP1. The results showed a same trend of above gene expression as the sequencing. The new data was provided in Supplemental Fig.3A.
8.The discussion aptly summarizes the findings and their implications. However, it should address potential limitations more thoroughly, such as the small sample size and the need for further validation in larger cohorts. The discussion on the potential clinical applications of BMP2 as a therapeutic target for AF is interesting. However, more emphasis on the challenges and future directions for translating these findings into clinical practice is needed.
Response: Thank you for this important comment. Due to the timeframe of the revision, we were unable to examine the level of BMP2 protein with a larger size of human AF atrial samples. We think that further studies with more samples are needed to confirm the change of BMP2 in AF patients. Besides, it is better to explore the role of BMP2R and its associated pathway in AF and how activated NLRP3 signaling regulates SMAD pathway. We have updated the discussions accordingly.
(Page 11, Line 356-359)
‘Besides, our study has a small number of human AF atrial samples to examine the BMP2 and its receptor expression. Further studies with larger sample sizes are needed to confirm the change of BMP2 in AF patients and clarify its correlation with AF progress.’
(Page 11, Line 365-367)
‘Moreover, it is needed to examine the role of BMP2R on the activated inflammation in atrial fibroblasts and provide insight to its potential mechanism related to NLRP3 signaling pathway.’
9.Figures and Tables
Figures are well-presented and provide clear visual representation of the data. Ensuring that all figures have comprehensive legends and are referred to appropriately in the text will improve clarity.
Response: We would like to thank you for this comment. During this revision, we provide more detailed figure legends to clarify the method, sample number and sample information to improve clarity. Please see the label changes in Figure legend section.
Tables summarizing patient demographics and experimental conditions are useful. Adding a table with key RNA-seq findings could also be beneficial.
Response: Thank you for this suggestion. We uploaded the key differential genes of RNA-seq data in the Supplemental table 3.
Reviewer 3 Report
Comments and Suggestions for Authors
I read with great interest the paper entitled „BMP2 diminishes angiotensin II-induced atrial fibrillation by inhibiting NLRP3 inflammasome signaling in atrial fibroblasts”.
The study presented in the manuscript explores the role of Bone Morphogenic Protein 2 (BMP2) in mitigating atrial fibrosis and subsequent atrial fibrillation (AF) through the inhibition of the NLRP3 inflammasome pathway. The authors employ both in vivo and in vitro models to elucidate the mechanistic pathways involved, providing a comprehensive analysis of BMP2's potential therapeutic effects against AF.
The investigation into BMP2's anti-fibrogenic properties and its modulation of the NLRP3 inflammasome in atrial fibroblasts provides novel insights into potential therapeutic targets for AF.
The results indicate a significant upregulation of BMP2 in AF models, which is associated with a reduction in NLRP3 inflammasome activation and atrial fibrosis. This suggests a protective role for BMP2 in AF pathogenesis, potentially offering a new avenue for therapeutic intervention.
The findings are significant, offering new insights into the modulation of atrial fibrosis and AF susceptibility. The study's methodology is sound, and the results are presented with clarity and rigor. The limitations discussed do not detract from the overall contribution of the work, which is a valuable addition to the field of cardiovascular research.
I would suggest adding a study design diagram in order for readers to understand easier how the study was designed.
Comments on the Quality of English LanguageThe manuscript is generally well-written. Only minor editing of English language is needed:
- line 45: the noun ”substrates” should be ”substrate”;
- line 84-86: the paragraph needs to be reformulated to enhance clarity.
Author Response
Response: We would like to thank you for summarizing and highlighting the main findings of our manuscript. We also appreciate the insightful suggestions for all your comments. The point-to-point response is as below to provide detailed updates of the revised manuscript.
I would suggest adding a study design diagram in order for readers to understand easier how the study was designed.
Response: Thank you for this suggestion. During this revision, we updated a ‘Graphic abstract’ figure to reflect our main findings. Please see the new figure in ‘Graphic figure’ section.
The manuscript is generally well-written. Only minor editing of English language is needed:
- line 45: the noun ”substrates” should be ”substrate”;
- line 84-86: the paragraph needs to be reformulated to enhance clarity.
Response: Thank you for pointing out this. We carefully review the spelling and words and corrected all issues to make the manuscript for easily reading.